# Surface frustration re-patterning underlies the structural landscape and evolvability of fungal orphan candidate effectors

**Mark C. Derbyshire**[1] **& Sylvain Raffaele** [2] ✉

Pathogens secrete effector proteins to subvert host physiology and cause disease. Effectors are engaged in a molecular arms race with the host resulting in conflicting evolutionary constraints to manipulate host cells without triggering immune responses. The molecular mechanisms allowing effectors to be at the same time robust and evolvable remain largely enigmatic. Here, we show that 62 conserved structure-related families encompass the majority of fungal orphan effector candidates in the Pezizomycotina subphylum. These effectors diversified through changes in patterns of thermodynamic frustration at surface residues. The underlying mutations tended to increase the robustness of the overall effector protein structure while switching potential binding interfaces. This mechanism could explain how conserved effector families maintained biological activity over long evolutionary timespans in different host environments and provides a model for the emergence of sequence-unrelated effector families with conserved structures.

Pathogens cause disease by manipulating host cell physiology to create a niche favorable to invasive growth and the completion of their life cycles. To do so, pathogens secrete small effector molecules active on host cells[1,2]. Pathogen effectors fall into diverse molecular classes. Secondary metabolites such as fumonisins and AAL toxins are produced by the fungal pathogens *Fusarium* and *Alternaria* to disrupt host sphingolipid metabolism and trigger cell death[3]. The gray mould fungus *Botrytis cinerea* uses transposon-derived small RNAs to interfere with the expression of plant defense genes[4]. Several fungal toxins are cyclic peptides post-translationally modified, such as *Cochliobolius* victorin that requires a specific plant protein target to confer virulence[5]. Many effectors are small secreted proteins with very diverse activities, among which are suppressing host immunity, hijacking nutrients, masking the pathogen presence, interfering with host development, and killing host cells[6–8].

Some pathogen effectors are active in the host cell while others function in the intercellular space[9]. Their activity often involves molecular interactions with host target proteins. For instance, the small cysteine rich effector Avr2 of the fungus *Cladosporium fulvum* acts as a protease inhibitor targeting the tomato protease Pip1 to block

its activity, which contributes to plant defense[10]. Resistant host genotypes harbor receptor proteins triggering defense responses upon binding to pathogen effectors or detecting the product of their activity. In tomato, the receptor-like protein Cf-2 recognizes Avr2 bound to Rcr3, a decoy protease closely related to Pip1, resulting in hypersensitive response and resistance[10]. The molecular interactions drive strong selective constraints on effector genes to manipulate their host targets efficiently while evading immune recognition by the host.

Selective constraints on effector genes left signatures in the genome of eukaryotic pathogens[11,12]. Typical effector genes evolve fast, showing frequent duplications, gain and loss, and a high degree of sequence divergence[13]. They often reside in repeat-rich genome compartments thought to accommodate high plasticity with limited deleterious fitness effects[14]. Fast effector gene evolution leads to expanded effector repertoires of hundreds of candidate effectors in the genomes of many filamentous microbes[15]. Effector candidates are often encoded by orphan genes that lack detectable sequence homologs outside their species or lineage, and therefore showing a patchy phylogenetic distribution. Effectors are typically small secreted proteins (SSPs) harboring an N-terminal secretion signal and several

[1]Centre for Crop and Disease Management, School of Molecular and Life Sciences, Curtin University, Perth, Australia. [2]Laboratoire des Interactions Plantes Micro-organismes Environnement (LIPME), INRAE, CNRS, Université de Toulouse, 31326 Castanet-Tolosan, France. ✉e-mail: sylvain.raffaele@inrae.fr

cysteine residues. In fungi, >95% of SSPs lack known functional domains[16]. This makes the identification of effector candidates, the inference of their evolutionary origin and their biological and molecular function very challenging.

Using knockouts in filamentous plant pathogens, several studies have provided evidence that individual effectors contribute to virulence. While the genome of the oomycete pathogen *Phytophthora infestans* harbors >500 effector genes of the RXLR type, silencing of the single Avr3a impairs virulence[17,18]. Remarkably, several mutants of the smut fungus *Ustilago maydis* were strongly reduced in virulence and the corresponding effectors showed diverse activities in plant cells[19,20]. More often, however, the deletion of single effectors leads to minor or no reduction in virulence[21]. The simultaneous deletion of multiple effectors is required to significantly reduce the virulence of the bacterial pathogen *Pseudomonas syringae* pv. *tomato*[22] and the fungus *B. cinerea*[23], highlighting some apparent degree of redundancy in the repertoire of pathogen effectors. In *B. cinerea*, which infects hundreds of plant species, the effect of multiple knockout on virulence differed markedly according to host species and tissues inoculated. This supports the view that the activity of a set of effectors depends on the repertoire of their targets[23]. Yet, the number of effector genes encoded by pathogen genomes does not increase with the range of hosts they infect[16], suggesting that some effectors are active on multiple host species, possibly on multiple targets. One example is *P. infestans* Avr3a that contributes to virulence through its interaction with plant E3 ligase, GTPase dynamin-related protein, cinnamyl alcohol dehydrogenase and perhaps other nuclear proteins[6].

Another layer of complexity emerges from interference between the activity of several effectors, either through epistatic interactions[24] or by cross-suppression of immune recognition[25,26]. Finally, effectors may promote virulence not by directly targeting host processes but rather by altering the host microbiota[27,28]. In summary, functional studies of effectors revealed that the simple equation "one effector, one biological activity" is generally false. Many pathogen effectors are promiscuous (active on multiple targets with no systematic fitness effect) or multifunctional (their activity on multiple targets confers an adaptive advantage).

Structural analysis revealed another level of redundancy and multifunctionality in the repertoire of filamentous pathogen effectors. The structure of several oomycete RXLR effectors uncovered a common four-helix bundle, folding around a WY domain[29]. This fold was proposed to serve as a structural scaffold allowing robust function and surface plasticity, enabling WY domain effectors to bind to diverse host proteins[30]. The structure of *Magnaporthe oryzae* AVR1-CO39 and AVR-Pia effectors exhibit a typical six-stranded β-sandwich fold analogous to *Pyrenophora tritici-repentis* ToxB necrotrophic effector. These effectors contribute to virulence on rice and wheat respectively and define the *Magnaporthe* Avrs and ToxB-like (MAX) fold[31]. Effectors from several *Blumeria* fungal species exhibit a structure analogous to RNAse-Like Proteins associated with Haustoria (RALPH)[32]. *Fusarium oxysporum* f. sp. *lycopersici* (*Fol*) effector Avr2/SIX3 shows a seven-stranded β-sandwich fold analogous to *Pyrenophora tritici-repentis* ToxA[33]. *Fol* Avr1/SIX4 and Avr3/SIX1 display a similar Fol dual-domain (FOLD) structure[34]. The AvrLm3, AvrLm5-9 and AvrLm4-7 effectors of *Leptosphaeria maculans* share the *Leptosphaeria* Avirulence and Supressing (LARS) fold with candidate effectors from 13 fungal species[26]. Structural predictions used to identify candidate effectors in the secretome of fungal pathogens revealed additional protein folds conserved across species[35-38].

Remote homology clustering allowed grouping 6538 fungal proteins related to effectors into 80 clusters spanning multiple pathogen species[39]. The structural genomics analysis of the secretomes of 21 fungi identified 485 structure-related families of five or more proteins, 374 of which were previously unknown[38]. Families containing known effectors had members from multiple pathogen and non-pathogen species, suggesting that apparently novel effector groups could have originated from conserved fungal proteins through duplication and sequence divergence. Strikingly, none of the structure-related families had been detected by classical sequence similarity searches alone, indicating a high degree of sequence divergence and highlighting that their structures are robust to heavy mutational burden. The best characterized WY and MAX folds revealed that structural similarity does not imply effector functional similarity[40]. *M. oryzae* MAX effector AVR-Pik binds and stabilizes rice susceptibility proteins with heavy metal-associated (HMA) domains[41], *M.oryzae* MAX effector AvrPiz-t binds to a RING E3 ubiquitin ligase to suppress rice immunity[42] and *P. tritici-repentis* MAX effector ToxB triggers host cell-death and susceptibility in wheat carrying sensitive *Tsc2* alleles[43]. Conserved effector structures therefore enable a high degree of molecular promiscuity and evolvability. Robustness to mutation and the capacity to acquire new functions are not directly constrained by selection for protein function, and vary among proteins with nearly identical structure[44]. Whether a specific evolutionary trajectory promotes the acquisition of robustness and evolvability in pathogen effectors is enigmatic.

Orphans are a particular class of protein for which no sequence homology with characterized proteins has been detected, indicative of extreme sequence divergence or recent gene birth. The extent to which orphans in fungal secretomes adopt similar protein structures and the evolutionary properties of such structures remain elusive. In this work, we deployed computational structural phylogenomics on ~5,000 secreted proteins from 20 fungal species and structural similarity network analysis to reveal the structural landscape of orphan candidate effectors (OCEs) at the entire subphylum level. Sixty-two structure-related families aggregated a majority of OCEs and were widely represented among species with diverse lifestyles and host ranges. Subsets of OCE families showed deep homology indicative of a distant common ancestry, with sequence and structural variation targeting mostly surface-exposed residues. Using in silico mutation scans, we identified clusters of co-evolving surface residues with contrasting effects on the overall fold robustness. Ancestral structure reconstruction revealed that fluctuations in local frustration of surface residue during OCE evolution associate with mutational robustness of conserved OCE families. We propose that re-patterning of surface frustration promotes the emergence of effector families that are both robust and highly evolvable.

## Results

### Ancient generic folds dominate the structural landscape of orphan candidate effectors

To explore the structural landscape of fungal orphan candidate effectors (OCEs), we predicted the complete repertoire of 3D structures for OCEs in twenty fungal species covering the phylogenetic and lifestyle diversity of the Pezizomycotina (Fig. 1A, B). Starting from complete genome sequences and the corresponding predicted proteomes, our pipeline selected proteins (i) including a predicted secretion signal, (ii) of mature size shorter than 300 amino-acids, (iii) lacking recognized PFAM protein domains, and (iv) with intrinsic disordered regions covering less than 50% of the sequence (Fig. 1A, B). From an initial set of 227,823 predicted proteins, we selected 4987 Orphan Candidate Effectors (OCE) for structure prediction using AlphaFold2[45]. We obtained 3927 OCE structures of average pLDDT score higher than 50. In subsequent analyses, we focused on the best resolved structures to reach a mean pLDDT>80 for 99.67% models reported in this work (Supplementary Fig. 1).

To classify orphan effector predicted structures according to similarity, we performed pairwise comparisons with 3911 OCE structures in DALI[46]. 477 proteins (12.2%) had no significant structural similarity (Z < 2) to any other OCE in the dataset. For a more conservative estimate, we focused on the 2561 orphan effectors with at least three analogs at $Z \geq 5.2$, leaving out 1350 (34.5%) singletons. With

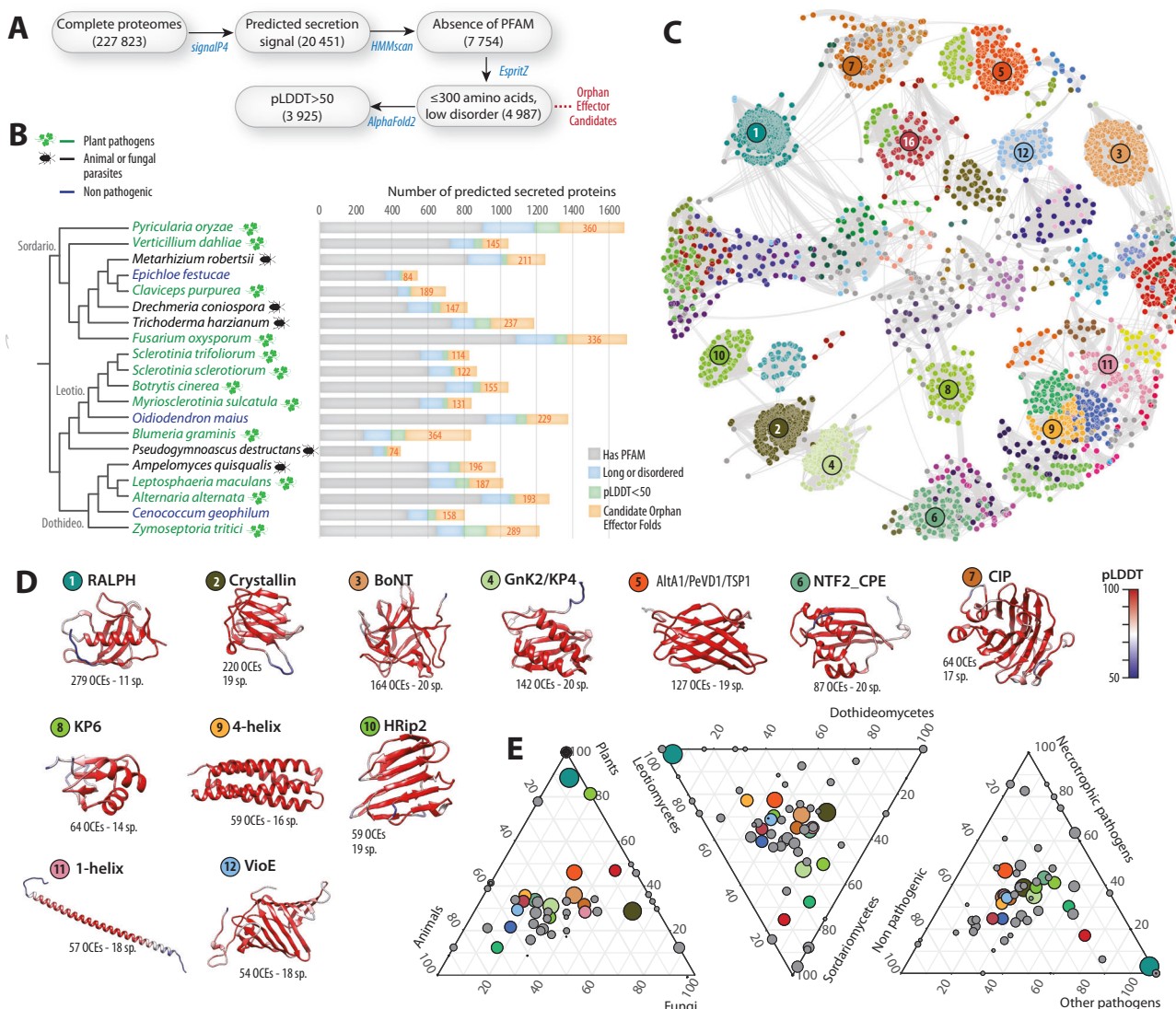

**Fig. 1 | Structural relationship between orphan candidate effectors in 20 fungal genomes. A** Bioinformatics pipeline for the systematic analysis of structures of orphan candidate effectors (OCEs) in fungal genomes. Filtering and analysis steps are shown as gray boxes with the number of retained proteins indicated between brackets. Analytical tools are indicated in blue. Out of 227,823 predicted proteins, 20,451 had a secretion signal, 4987 we selected for structure prediction, 3925 had a reliable structure predicted. **B** Fungal species selected to cover the phylogenetic and lifestyle diversity of the Pezizomycotina. For each proteome, the distribution of secreted proteins with different characteristics based on the pipeline is indicated as a histogram. **C** OCE structural similarity network with vertices colored according to

detected communities forming structure-related families. The 12 major families including ≥54 proteins are numbered by decreasing protein count.
**D** Representative predicted structures corresponding to the 12 major families. Numbered and colored stickers refer to communities shown in (**C**). Residues are colored according to their pLDDT score. The number of OCEs and species in which they occur (sp.) is indicated. **E** Contribution of fungal species classified by lineage, host type and lifestyle to each of the 62 major families. Each circle represents one OCE fold, sized according to the number of proteins it contains. The 12 major families are colored as in (**D**). Source data are provided as a Source data file.

those, we built a structural similarity network featuring OCE predicted structures as vertices and DALI Z-scores as weighted edges (Fig. 1C). The structural similarity network featured nodes of high connectivity, corresponding to protein folds shared between multiple orphan effectors. In agreement, the network has a high clustering coefficient (0.86) and high heterogeneity (0.579) but low density (0.195) and low centralization (0.35).

Six effectors from the network had experimentally determined structures (Supplementary Fig. 2). Consistent with previous reports[38], the match between experimentally determined structures and Alpha-Fold2 predictions was very good, with RMSD ranging from 0.468 to 3.94 angstroms. To characterize OCE structure-related families, we performed community detection on the OCE structure similarity network and compared OCE predicted structures to those in the PDB database. Combining hierarchical community detection with HiDeF

and manual curation, we identified 62 families of at least five effector candidates, representing 2371 OCEs across the twenty fungal genomes (60.4% of all OCE structures predicted with pLDDT>50, Fig. 1C). The most abundant was the RALPH family found in 279 OCEs from 11 species, among which 262 were from the powdery mildew pathogen *Blumeria graminis* f. sp. *hordei* (Fig. 1C, D). The most widespread families, retrieved in all 20 species, were related to toxin folds: the β-trefoil found in *Clostridium* neurotoxins and plant ricins, the β-(α-β$_2$)$_2$ sandwich of KP4 fungal killer toxin and plant ginkbilobin-2, the α-β$_4$ core sandwich of the nuclear transport factor 2 and *Clostridium perfringens* enterotoxin. Eighteen families were detected in over 80% of the species (Fig. 1C, D, Supplementary Figs. 3, 4), including additional toxin folds such as the β$_8$ double Greek key of βγ-crystallin and the yeast killer toxin WmKT[47], and the (α-β-α)$_2$ sandwich of the *Ustilago maydis* KP6 killer toxin[48]. Among the eighteen broadly conserved

families, we identified five previously associated with known fungal effectors (Fig. 1D): RALPH[49], Alt-A1[50–52], HRip2[53], and the 5-helix bundle found in *Drechmeria coniospora* g7941 effector (6zpp). We identified three additional known effector families (Fig. 1D): MAX (44 OCEs in 3 species)[31], Avr2/ToxA (39 OCEs in 6 species)[33] and Tox3 (6 OCEs in 5 species)[54].

Out of 62 major OCE families, only three were specific to a single fungal species, suggesting that specialization in these OCEs is limited. To support this claim, we analyzed the distribution of the 62 families across fungal lineages, host group and lifestyle (Fig. 1E). Fourty four OCE families were detected in Leotiomycetes, Sordariomycetes and Dothideomycetes with cases of lineage-specific enrichments (e.g MAX and Tox3 folds enriched in the Sordariomycetes). Only five OCE families were restricted to a single lineage. Thirty-five OCE families were detected in pathogens infecting plant, fungal and animal hosts, while 13 OCE families were restricted to fungi infecting only one of these host groups. Twelve families were restricted to plant pathogens including MAX and SIX3. Forty-six OCE families were detected in pathogens with a mutualistic, a necrotrophic and other pathogenic lifestyles. Apart from the KP6 family, which was absent from animal pathogens, all the most abundant OCE families were detected in fungi from all lineages, infecting all host groups and with all lifestyles.

In agreement with previous analyses of fungal secretomes[38,39], we highlight a few OCE families highly represented at the subphylum level, consistent with deep common ancestry. Although considered as orphans based on classical sequence homology searches, many OCEs adopt effector and toxin folds described previously and distributed broadly across fungal species with various affinities and lifestyles. This suggests that ancient OCE folds remained adaptive in diverse biological contexts, and in spite of considerable sequence divergence.

### OCE families display a stable core with flexible surface-exposed regions

The detection of sequence-unrelated OCEs with similar 3D structures suggests that modern OCEs derived from ancestral structures which remained stable over evolution, in spite of extensive sequence divergence[31,38]. To document the extant diversity in major OCE families, we first compared sequence-based evolutionary distance to structural similarity (Fig. 2A, Supplementary Fig. 5). We performed the same analysis for Histone 2A from the twenty fungal species used in our analysis to serve as a control. For all protein groups, structural similarity decreased regularly for evolutionary distance 0 to 200, and tended to stabilize at Z-score -5.0 beyond this distance. The number of proteins showing unexpectedly high structural similarity to other members of the family (above 97.5% prediction interval) was 0 for Histone 2A, 68 for Alt-A1, and 89 for KP4 toxins. Among OCE families, the proportion of proteins with unexpectedly high structural similarity to other members ranged from 80.4% (Crystallin) to 23.4% (KP6). OCE families therefore maintain a high structural similarity in spite of high evolutionary distance.

In some effector families, variable residues tend to reside at the surface of proteins[11]. To test whether structural similarity related to amino acid burial in OCEs, we analyzed structure-based alignments of OCEs representative of sequence diversity in Alt-A1 and BoNT families (Fig. 2B–D). Although some of the selected OCEs shared as little as 3% identity with each other, their structures superimposed with good accuracy. Residues with the highest relative surface exposure localized to regions where the structural alignments decreased in quality (Fig. 2C, Supplementary Movies 1, 2). We observed a global correlation between relative surface exposure and alignment deviation (RMSD, in Angstroms) for aligned residues, supporting a link between conformational flexibility and surface exposure (Fig. 2D). We also noted a tendency for more conserved residues to reside in buried, structurally stable regions, although the high sequence divergence among the

selected representative sequences did not allow to asses this trend accurately (Fig. 2D).

To analyze evolutionary relationships within OCE families, we used sensitive all-vs-all profile hidden Markov model (HMM) comparisons for all fungal BLASTp homologs in the NCBI nr database of 11 of the major OCE families (Fig. 2E, Supplementary Fig. 6). In the Alt-A1 family, we clustered 11,251 amino acid sequences into 494 groups of proteins with >30% AA identity, and further grouped these into 13 super-clusters based on HMM comparisons. OCEs from our 20-genomes analysis appeared in all 13 super-clusters. Super-clusters 1 to 7 showed weak homology relationships, supporting the view that ancient and broadly conserved protein folds were recruited as effectors in fungi with diverse lifestyles. Though there was structural similarity between OCEs and metazoan and bacterial proteins, we did not detect evidence of common ancestry between super-clusters 1 to 7 and 8 to 13. Therefore, although a single ancestral group dominates the structural family, we cannot rule out the possibility of convergence in structure of unrelated sequences. Similar conclusions could be drawn from analysis of the other 10 structural families, supporting the existence of OCE families identified based on structural similarity analyses. For eight of the 11 OCE families, at least 75% of the predicted effectors had remote sequence homology to structural hits from fungi in PDB (Supplementary Fig. 6). Over all, for homologs of the Alt-A1, BoNT, Gink2/KP4, HRip2, KP6, and NFT2/CPE OCE families, the largest super-clusters comprised 89%, 94%, 91%, 97%, 98%, 69%, and 89%, respectively, of HMMs and singletons (Supplementary Fig. 7). Whereas HMMs comprising homologs of OCEs from the atg, CIP, g7941, SIX3, and VioE families were spread more evenly across super-clusters, with the largest super-clusters comprising from 35% (SIX3) to 47% (g7941) of HMMs and singletons.

We conclude that major OCEs likely derive from ancient protein folds that accumulated surface variation and maintained a stable core despite extensive sequence divergence.

### Modern OCEs accumulated co-selected mutations reducing structural divergence

To test the impact of mutations on major OCE families, we reconstructed ancestral OCE structures for clades of KP6 and Alt-A1 families (Fig. 3A). To highlight structural regions prone to sequence variation, we mapped the positions of modern variants from each clade on the ancestral protein structure (Fig. 3B, Supplementary Movies 3, 4). Unexpectedly, multiple weakly conserved residues occupied the core of the ancestral OCE structure, and amino acid conservation did not clearly discriminate between core and surface regions. To determine the extent to which these amino acid changes were selected independently, we calculated the frequency of mutation co-occurrence in modern OCEs. We shuffled extant variable residues 10,000 times to estimate a null probability of mutation co-occurrence and estimate co-selection *p*-values for observed mutation pairs. In the KP6 clade, 64 residues (84.2% of variable positions) had co-selection p-val <1.0E−10. In the Alt-A1 clade, 60 residues (38.7% of variable positions) had co-selection p-val <1.0E−10. Significant co-mutation relationships defined 6 groups of co-selected mutations both in KP6 and Alt-A1 (Fig. 3C, Supplementary Movies 5, 6). Apart from co-selected cluster 2 in KP6 (red, Fig. 3C), co-selected amino acids tended to distribute across the whole OCE structure rather than localize to specific sites of the protein.

To assess the effect of extant sequence diversity on OCE structural diversity, we compared structures predicted for extant OCEs with the structure of their common ancestor. The self-alignment of the ancestral structure served as a reference to estimate the structural divergence (SD) induced by natural diversity. In the KP6 clade, most extant variants had SD < 2.0, indicative of a very mild change in structure, while in the Alt-A1 clade, median SD was 4.3 (Fig. 3D). To estimate the relative contribution of individual mutations to OCE structural

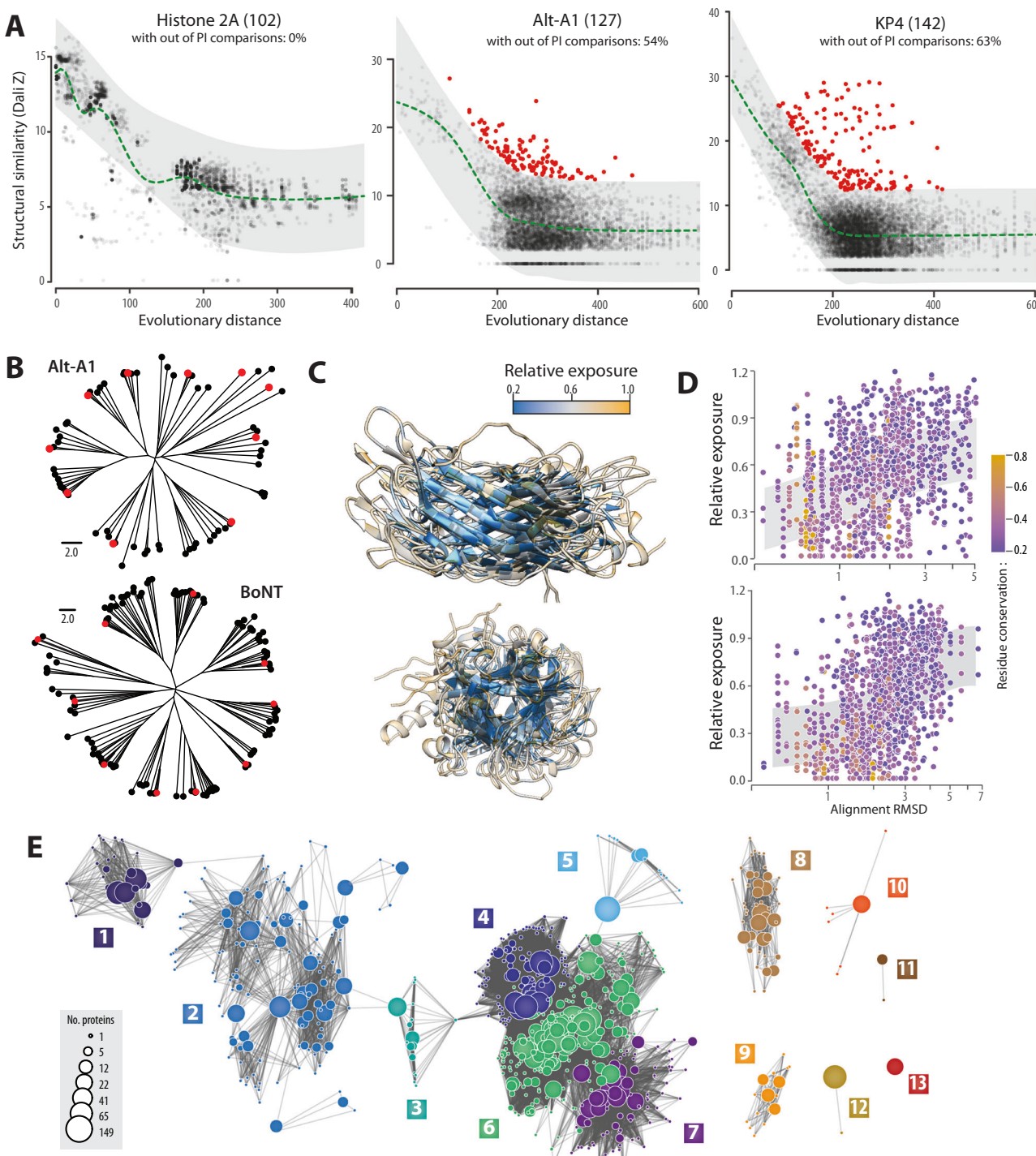

**Fig. 2 | The extant sequence and structural diversity in major OCE families reveal stable cores with flexible surface-exposed regions. A** Structural similarity (Dali Z-score) according to evolutionary distance (Jukes-Cantor corrected distance calculated on curated sequence alignments) for histone 2A and the OCE families Alt-A1 and KP4 toxins. The green dotted lines are the local average. Pairwise structure comparisons falling outside of the 97.5% prediction interval (PI, gray ribbons) are shown as red dots. Numbers in brackets indicate the number of protein structures compared. The percent of proteins with Z values above the predicted interval is indicated for each family. **B** Structure similarity neighbour-joining trees for members of the Alt-A1 and BoNT families, with representative structures shown as red circles. **C** Structural alignment of the corresponding representative proteins colored according to the residue relative exposure. **D** Relative residue exposure according to local structural alignment accuracy (RMSD, in Angstrom). Gray ribbons show LOESS regression-centered 50% prediction intervals. Dots are colored according to residue conservation. **E** The overall diversity of the Alt-A1 family obtained through sensitive HMM comparisons of homologs from the NCBI nr database. Each circle corresponds to a cluster of 1 to 169 sequences, with homology relationships shows as edges. The 13 superclusters are numbered and shown in different colors. Superclusters 1 to 7 are connected by homology to a single member. Source data are provided as a Source data file.

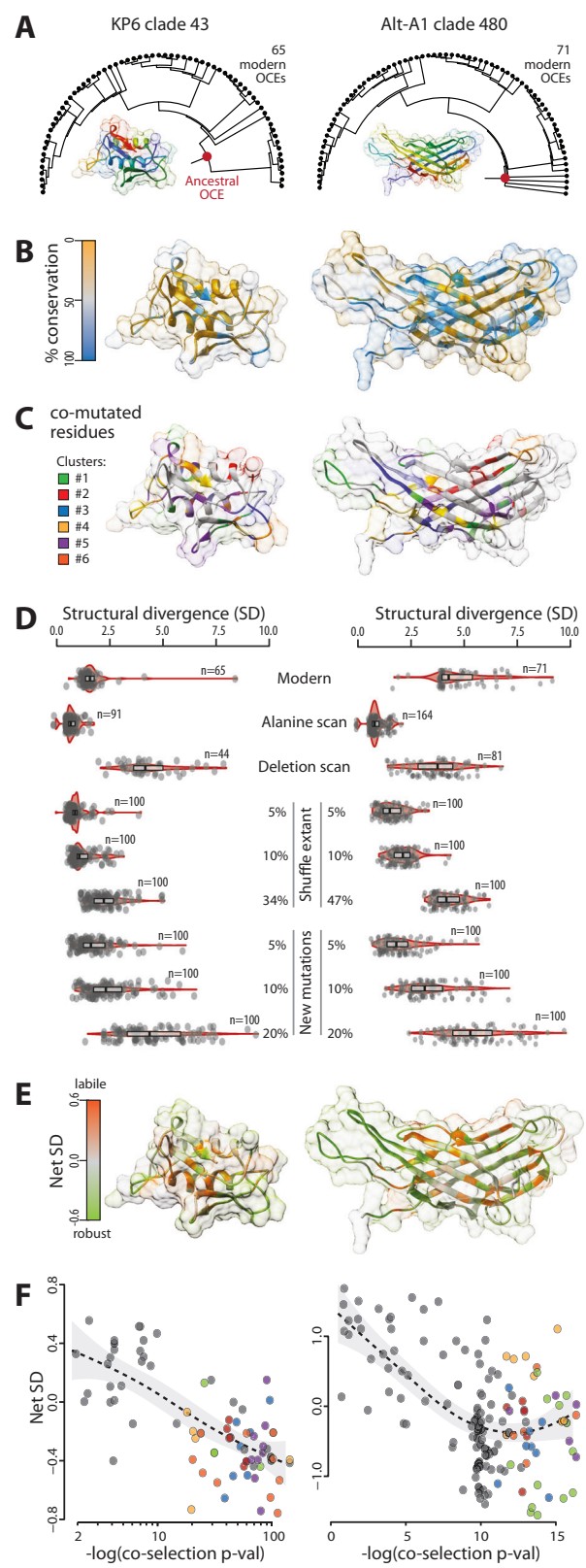

**Fig. 3 | Mapping of robustness-increasing mutations based on natural diversity and in silico mutation scans in clades of KP6 and Alt-A1 OCEs. A** Reconstruction of OCE ancestral structure in KP6 clade 43 and Alt-A1 clade 480. The phylogenetic trees used for ancestral sequence reconstruction show modern OCEs as black circles and ancestral nodes used in structure prediction as red circles. The Alphafold models of the common ancestors are shown in rainbow colors. **B** Sequence conservation (%) in modern OCES mapped onto the protein structure of their common ancestor. **C** Mapping of clusters of residues significantly co-mutated onto ancestral protein structures. **D** Structural divergence (SD) caused by extant natural variants ('Modern') and artificial mutations in ancestral OCEs. Structural divergence between variant OCEs is expressed relative to ancestral OCE self-alignment score. Boxplots show median values (thick line), first and third quartile values (box) and 1.5 times the interquartile range (whiskers). **E** Mapping of residues net SD effect onto the structure of the ancestral OCE. Net SD corresponds to the difference between SD obtained after the mutation/deletion of individual residues and SD obtained after multiple simultaneous mutations. **F** Relationship between residues SD effect (Y-axis) and their probability of co-selection (X-axis, as log10 of $p$-value). Residues are colored as in (**C**) according to the cluster they belong to. The dotted line shows local average, the gray area corresponds to the 95% confidence interval. Source data are provided as a Source data file.

existing variants and introduced multiple new mutations randomly to modify simultaneously 5% to 47% of the ancestral sequence and assess the SD effect of multiple mutations on the ancestral protein structure (Fig. 3D). All Alt-A1 and 97% of KP6 multiple mutants had a SD smaller than the sum of SD for individual mutated residues they contain, indicating that the effect of individual mutations on SD is not strictly additive. We hypothesized that the missing SD in multiple mutants was due to compensatory effects contributing to the robustness of OCE folds. We used this information to derive the net SD effect of mutations at each position (see methods). This approach identified 47 residues (51%) in KP6 and 92 residues (56%) in Alt-A1 with negative net SD effect that could increase the mutational robustness of OCEs. Robustness-increasing residues tended to reside away from the OCE structural core (Fig. 3E, Supplementary Movies 7, 8).

In our hypothesis, mutations at robustness-increasing positions would compensate for structural variation due to mutations occurring elsewhere in the protein, maintaining a conserved structure despite sequence divergence. To support this model, we analyzed the SD effect according to the likelihood of co-selection in KP6 and Alt-A1 clades (Fig. 3F). On average, the SD effect decreased with probability of co-selection in KP6 and in Alt-A1. Within clusters of co-selected residues, the amplitude of the net SD effect covered 17 to 49% of the total amplitude in the corresponding protein, supporting the view that robustness-increasing residues co-evolved with residues with a strong SD effect.

## OCEs evolved through oscillations in surface residue frustration

Proteins acquiring mutations enhancing stability are more likely to tolerate destabilizing mutations, which may confer functional benefits[55,56]. Indeed, surface residues showing high frustration, corresponding to unfavorable interactions in the protein native state, often contribute to binding interactions releasing frustration[57]. To test whether OCEs exhibit a frustrated and evolvable surface surrounding a thermodynamically stable core, we assessed the relationship between frustration index[58] and relative surface exposure. Across the three families we tested (AltA1, KP6, and Gink2/KP4), frustration index decreased (corresponding to higher levels of frustration) with residue surface exposure (Spearman's rho = −0.4, Fig. 4A). Highly frustrated amino acids had a significantly higher average relative surface exposure of 0.66 compared with 0.52 for minimally and neutrally frustrated residues ($P = 2.2e−16$, Fig. 4B, Supplementary Fig. 8). To identify OCE regions prone to frustration variation, we calculated frustration index variance in four clades of four different families, and mapped these

divergence and document the theoretical sequence diversity that the major OCE folds can withstand, we performed mutation scans followed by structure prediction and comparison on the KP6 and Alt-A1 ancestral proteins. We mutated individual amino acids to Alanine and used a sliding window deletion scan to estimate SD for individual residues in the ancestral KP6 and Alt-A1 proteins. Next, we shuffled

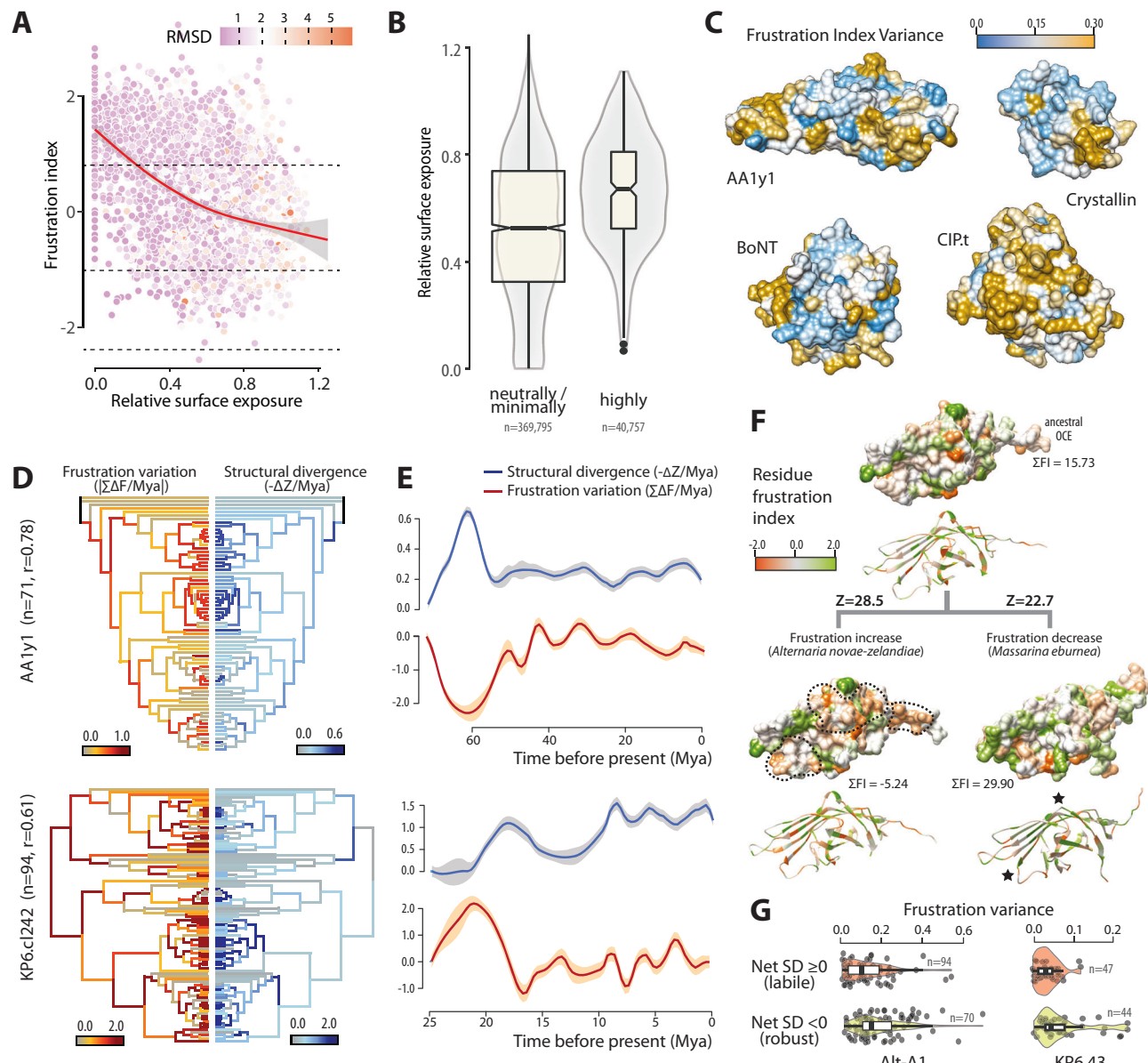

**Fig. 4 | Re-patterning of surface residue frustration buffers structural variation in OCEs. A** Frustration index negatively correlates with relative surface exposure in Alt-A1, KP6, and GNK2/KP4 OCE families. Frustration index decreases with the degree of residue frustration. The red line shows loess regression, gray area is the 90% confidence interval, dotted lines delimit minimally (0.78) and highly (−1.0) frustrated thresholds. Data points are colored according to Cα structural deviation from predicted ancestral structure (RMSD, root mean square deviation, in Angstroms). **B** Comparison of relative surface exposure for residue classified as neutrally/minimally frustrated and highly frustrated in Alt-A1, KP6, and GNK2/KP4 OCE families. Boxplots show median values (thick line), first and third quartile values (box) and 1.5 times the interquartile range (whiskers). **C** Frustration index variance in clades from four OCE families mapped on the structure of the reconstructed clade ancestor. **D** Frustration variation (red, left side) and structure divergence (SD, blue, right side) rates during the evolution of OCEs from an Alt-A1 and a KP6 clade mapped on time-calibrated phylogenies. *N* is the number of modern OCEs per

group, r is Pearson's product-moment correlation between frustration variation and structural divergence across all branches of the tree. **E** Frustration variation (red) and structural divergence (blue) over time across every possible evolutionary path in trees shown in (**D**). Lines show mean value from a loess regression, shaded areas are the 95% confidence interval. **F** Reconstructed evolution of structure and residue frustration in two lineages of members of the Alt-A1 family. Each protein structure is shown as molecular surface (top) and ribbon diagram (bottom), with the reconstructed ancestral protein at the top of the panel and two of its modern descendants at the bottom. Z, structural similarity score; ΣFI, sum of frustration index for all residues in the protein. Black dotted lines delimit patches of increased frustration, black stars indicate the position of highly variable loops. **G** Frustration variance for residues showing positive (green) and negative (red) SD effect in mutational scans of two OCE groups. Mya, Million years ago. Boxplots show median values (thick line), first and third quartile values (box) and 1.5 times the interquartile range (whiskers). Source data are provided as a Source data file.

values on protein structures (Fig. 4C). The four families showed multiple surface patches with high frustration variance that could mediate differential binding with diverse molecular partners.

To document the evolution of structure and frustration in major OCE families, we built maximum likelihood phylogenies, reconstructed ancestral protein structures, and analyzed frustration for 15 HMM sequence clusters covering six OCE families and representing

1307 modern OCEs and 1289 ancestral OCEs. Using time-calibrated trees, we estimated the rate of SD and amino-acid frustration along each branch of the phylogenies (Fig. 4D). We observed a general correlation between the rate of frustration variation and the rate of SD (median Pearson's r = 0.66) along branches, indicating that amino-acid thermodynamic frustration is a major driver of structural change in OCEs (Supplementary Figs. 9, 10). The path from root to tip passed

through several frustration and SD local maxima. To document the evolutionary dynamics of frustration and SD in selected OCE clades, we collected these metrics along every evolutionary path in trees, using estimated branching times as a clock (Fig. 4E). It revealed cycles of frustration maxima followed by frustration release. Frustration decrease roughly coincided with bursts of SD relative to previous nodes, indicating that the release of frustrated contacts is a major step in the structural evolution of OCEs. In some OCE clades, modern proteins close to their frustration maximum showed reduced structural divergence from the clade ancestor compared to modern proteins with low frustration (Fig. 4F). In agreement, residues with a robustness-increasing effect (net SD < 0) identified in Alt-A1 and KP6 clusters (Fig. 3E) respectively showed 1.5 and 2.0 times higher frustration variance than positions with a destabilizing effect (t test $p$-val = 1E−03 and 8E−04, respectively, Fig. 4G). These results suggest that mutations increasing OCE frustration contribute to sequence and functional diversity while maintaining the overall protein structure robust, leading to the emergence of large families of sequence unrelated, structurally conserved effector candidates.

## Discussion

Pathogen effectors are engaged in a molecular arms race with the host resulting in conflicting evolutionary constraints to subvert host physiology efficiently on the one hand, and avoid detection by the host immune system on the other hand. In this work, we show that conserved fungal effectors diversified through re-patterning of frustrated surface residues. The underlying mutations tend to increase the robustness of the overall effector protein structure while switching potential binding interfaces. This mechanism could explain how conserved effector families have maintained biological activity over long evolutionary timespans and in different host environments, and diversified exposed residues to escape host immune detection. The finding that effector proteins can withstand high mutational burden by building up surface frustration provides a model for the emergence of widely distributed families of sequence-unrelated effector families with analogous structures.

The limited structural information available to date suggests that some oomycete effector families adopt a conserved modular core with variable surface residues to accommodate evolutionary constraints[11,29,30]. Low sequence homology and the absence of distinctive motifs have prevented the large-scale detection of effector families in fungi. Recent structural studies and the advent of computational structural genomics have identified sequence-unrelated families of fungal effectors[29,31,36]. Focusing specifically on small orphan secreted proteins, we reveal that the majority of fungal secreted proteins with no recognizable domains, the secretome dark matter, is made of families of structurally analogous effector candidates. Most of these OCE families are widely conserved across fungi with diverse lifestyles. In these families, amino-acid and structure conservation decreased with relative surface exposure, generalizing the model of effectors having a stable core and a highly variable surface, and expanding the known repertoire of effector stable cores that tolerate extreme surface variation.

We focused on 62 effector families that make up the majority of the Ascomycete OCE repertoire. For the most widespread families, sensitive HMM searches identified distant homologs in hundreds of species[39,59], suggesting that effector families may have deep common ancestry. A few OCE families were lineage-specific or showed species-specific expansions. If OCE families were often represented by several members in each genome, massive expansions like in the case of oomycete RXLR and CRN effectors[17] were rather rare (e.g., RALPH-like in *B. graminis*, MAX in *P. oryzae* and *C. purpurea*), suggesting that segmental gene duplication may not be a major contributor to the evolution of fungal OCE families. Two major OCE families (GNK2/KP4 and KP6) showed structural similarity to yeast and filamentous fungi

killer toxins[60]. In strains of the smut fungus *Ustilago maydis*, KP4 and 6 are encoded by persistent dsRNA viruses, and secreted by the fungus to kill competing strains that do not contain cognate toxin resistance genes. The *U. maydis* killer phenotype is inherited by horizontal gene transfer (HGT)[61], and HGT to plants has been documented for KP4[62]. Furthermore, HGT is a key process in the inheritance of OCEs from the SIX and ToxA families[63,64], and in the emergence of pathogenicity in *Metarhizium* entomopathogens[65], suggesting that HGT may be an important process in the evolution of OCEs.

Based on the panel of Ascomycete species we selected, we estimate that ~70 families of structure-related OCEs cover the majority of the fungal uncharacterized effectome (Supplementary Fig. 11). Grouping into structural families can guide the functional characterization of pathogen effectors. A complete understanding of fungal pathogenicity will require expanding the current effort to include (i) other filamentous pathogen lineages such as Basidiomycete fungi and oomycetes (see, ref. 38), (ii) OCEs that were excluded from our pipeline such as those longer than 300 amino-acids, harboring PFAM domains or with no canonical secretion signal, (iii) OCEs with no clear analogs in protein structure databases and that did not group into structure-related families.

The wide distribution of many OCE families suggests that some could have generic functions in pathogenicity, such as broad host toxicity[66,67] or competition with other microbes[28,68]. In spite of sharing a similar structure, sequence conservation in OCE families is generally very low, which may be the result of evolution to escape host or competing microbe recognition. It also raises the possibility that OCEs from the same family may have different biological activities. This view is notably supported by differential phytotoxic activity in families of Necrosis- and ethylene-inducing peptide 1-like proteins[67,69] and other cell death-inducing proteins[70]. Our results suggest that mutation-driven increase in surface frustration promotes new molecular interactions and neofunctionalization without major alteration to the ancestral fold. Studies on intrinsically disordered proteins showed that frustration promotes promiscuous molecular interactions, leading to versatile protein complex assembly, instead of specific binding[71,72]. This property may be mediated by increased flexibility of surface loops[73]. Increase of frustration in OCEs may therefore be a mechanism to expand the repertoire of their binding targets, contributing to the emergence of general-purpose effectors. This study raises hypotheses regarding the evolution of effectors function that remain to be tested experimentally. A major objective for future work will include determining whether the energy landscape of effector proteins associates with specific binding activity and biological function. Molecular resurrection experiments will be required to test the evolutionary scenarios emerging from our analyses. The Alt-A1 OCE family, with members characterized from *B. cinerea*, *S. sclerotiorum*, *T. virens*, and *V. dahliae*[74] is an attractive target for such experiments.

The intramolecular interaction network of proteins allows them to accommodate substitutions to a certain extent, while still maintaining their native fold[75]. In large populations, as for many fungi and other microbes, proteins evolve greater stability, a biophysical property that is known to enhance both mutational robustness and evolvability[44]. Experimental evolution approaches indeed showed that the emergence of new functions was facilitated by the use of a more robust evolutionary starting point[72]. Our analysis of natural diversity, in silico mutation scan, and ancestral structure reconstruction indicate that several OCE folds are highly robust to mutations, notably through the ability to accumulate surface frustration. This property could therefore facilitate functional switches in OCEs.

The analysis of evolutionary paths in two OCE clades indicated that frustration tends to fluctuate over time. This suggests that the amount of frustration that OCE scaffolds can withstand is limited. Following the principle of minimal frustration, we would expect highly

frustrated OCEs to unfold more rapidly[76], and thus to be active for shorter time periods or in a more restricted range of conditions. Alternatively, highly frustrated OCEs may fall in kinetic traps leading to less efficient folding, or show an increased risk of forming deleterious aggregates[77]. Release of frustration during OCE evolution roughly corresponded to phases of structural divergence, creating cycles of frustration increase followed by structural re-arrangements. The tradeoff between structural stability and biological activity has been reported in diverse protein types and is a major bottleneck in protein engineering[78]. The evolution of fungal OCEs recapitulates the three solutions typically employed in protein engineering: (i) OCE families evolve from highly stable parental proteins with a robust core, (ii) structural destabilization during functional evolution is minimized by increasing surface frustration and the co-selection of residues with contrasted stabilizing effects, (iii) OCEs evolutionary paths establish a long-term balance between structure and thermodynamic stability. Patterns of thermodynamic stability loss and restoration were reported during the evolution of new enzyme activities[55], suggesting that fluctuations in surface frustration may correspond to functional switches in OCEs. These frustration patterns imply that the structural OCE template is not lost over evolution, including following neo-functionalization, and remains available for further diversification. It may also revert to a previous native state leading to apparent structural convergence.

Progress in effector structural biology and structural genomics enabled by deep learning algorithms accelerated the pace of the characterization of filamentous pathogen effectors. We propose here an innovative structural and phylogenomics pipeline combining ancestral structure reconstruction, mutation scans, and frustration analysis to investigate the molecular mechanisms and evolution of effector proteins. The structural phylogenomics strategy applied to OCE families of interest should help identifying residues involved in protein-protein interaction, infer probable functional switches during OCE evolution, infer thermodynamic properties of effectors and their evolutionary potential. Ongoing progress in the functional characterization of fungal secreted proteins will allow formally testing and refining mechanistic and evolutionary models derived from structural phylogenomics studies.

## Methods

### Identification of orphan candidate effectors and structure prediction

Complete predicted proteomes for 20 Ascomycete fungi were downloaded from the public repositories listed in Source Data. Secreted proteins were predicted using signalP-4.1g for Cygwin[79] which is recommended for pathogen effector predictions[80]. Signal peptides were trimmed and proteins longer than 300 amino acids were excluded from further analysis. Mature sequences of secreted proteins were searched for conserved domains with the hmmscan.pl script using the Pfam-A 35.0 database (dated 2021-11). Proteins with no Pfam-A hit of $p$-value < 0.01 were considered as orphans. Orphans were searched for intrinsic disorder regions using the EspritZ server version 1.3[81], with prediction type X-ray and 5%FPR decision threshold. Proteins with intrinsic disorder regions spanning >50% of their mature sequence were excluded from further analysis, leaving 4987 mature sequences as input for structure prediction. We chose AlphaFold2 to model OCE 3D structures since it has been rigorously tested and shown to offer currently the best performances for effectors of filamentous plant pathogens[38]. OCE structures were predicted using the ColabFold: AlphaFold2 w/MMseqs2 BATCH notebook with parameters msa_mode: MMseqs2 (UniRef+Environmental), num_models: 4, num_recycles: 3, stop_at_score: 100. Structures with average pLDDT score <50 were excluded from further analysis. Major OCE folds in our structural similarity network had an average pLDDT from 80.01 to 89.06, 99.67% of all OCE structures analyzed in this work had pLDDT>80.

### Structural relationships analysis and rendering

Structural similarity between the 3925 OCE structures was calculated using the all-versus-all function in DALILite 5.0, which showed top performances with the alignment of small protein domains as OCEs[82]. The resulting matrix of pairwise similarity Z-scores was converted into a structural similarity network with the igraph package in RStudio v1.4.1106, filtering out edges of weight (Z-score) < 5.2 and nodes with degree < 3 and using the Kamada-Kawai force-directed layout algorithm. The resulting network was exported as a.gml file using the plotCytoscapeGML function from the NetPathMiner v1.8.0 package. Similarity to known protein structures was assessed using DALI against a local instance of the PDB database as accessed on May 12, 2022[83]. Draft OCE structural groups were identified using community detection with the Louvain algorithm implemented in HiDEF[84] with maximum resolution = 50.0, consensus threshold = 75, persistent threshold = 5, target community number and weight column disabled. Final OCE structural groups were refined manually based on community detection and structural similarity with the PDB database. Groups were named according to the dominant match with PDB or the PDB hit with the highest Z-score, with priority given to proteins of known function. Structural similarity in major OCE families was further supported by TM-scores and RMSD calculated with TM-align[85] (Supplementary Fig. 12).

### Analysis of natural sequence and structure diversity

Structural similarity was assessed using the all-versus-all function in DALILite 5.0[46] and the TM-align program updated on April 12, 2022[85]. Evolutionary distance was based on multiple alignments generated with muscle[86] and manually curated to keep positions covered at 80% minimum, and assessed using the distmat program in EMBOSS[87] with Jukes-Cantor correction for multiple substitution. To generate structure similarity trees, Dali Z-scores were converted to distances by calculating the difference to the maximum Z-score in each matrix, and neighbor-joining trees were created with BioNJ. The longest branches were pruned in TreeGraph2[88]. Representative structures were aligned in UCSF Chimera version 1.11.2 build 41376[89] with MatchMaker for pairwise superposition with the Needlman-Wunsch algorithm and the PAM-250 matrix, secondary structure score (30%) included, secondary structure assignments computed and iteration by pruning long atom pairs until no pair exceeds 0.1 angstrom. To refine the superposition and retrieve the corresponding alignment, the Match->Align tool was used with residue-residue distance cutoff 5.0 angstroms, residue aligned in column if within cutoff of "at least one other", iterate superposition/alignment until convergence, superimpose full columns in stretches of at least 3 consecutive columns. For deep homology analysis, we used BLASTp searches of OCEs (and for Alt-A1 their top five PDB hits) against the NCBI nr database with e-value cutoff 0.05, word size 2, scored using the PAM30 matrix with gap costs 8 for existence and 1 for extension. Amino acid sequences were clustered with a mutual coverage of >=80% and a similarity of >=30% using MMseqs2 version 13.45111[90]. These clusters were then used to create alignments using Decipher version 2.24.0[91]. We then used hhsuite3 version 3.3.0[92] to create HMMs from MMseqs clusters and conduct an all vs all pairwise HMM-HMM comparisons and sequence-HMM comparisons between singletons and clusters. MMseqs clusters were assigned to super-clusters where all members of a super-cluster had a significant HMM similarity to at least one other member at e-value ≤ 1E−5 (i.e., the connected components of the whole graph).

### Mutation scans and determination of residue structural divergence effects

Ancestral OCE sequences were determined with GRASP version 2020.05.05[93] using the JTT evolutionary model. Sequence reconstructions at the most ancestral nodes were submitted to AlphaFold2 to obtain the ancestral OCE structures. The residue percent

conservation corresponds to the proportion of modern OCEs harboring the ancestral residue at a given position. A null model for mutation co-occurrence was estimated by randomly shuffling extant polymorphisms 10,000 times and counting the occurrence of all mutation pairs. P-values are the product of mutation pair occurrence under the null model exponent the number of observed mutation pairs, followed by Bonferroni correction for multiple testing. Groups of co-selected mutations were identified by community detection applied to the network of co-occurring mutations, using the Louvain algorithm implemented in HiDEF with default parameters. For Alanine scans, amino acids were turned into Alanine one by one. For deletion scans, five consecutive amino acids were deleted in a sliding window approach of step two. For variant shuffling, a repertoire of all variants at a given position detected in the OCE clade was built, these variants were introduced randomly in the ancestral sequence to obtain 100 sequences with an average 5 to 47% positions mutated. For new mutation scans, a repertoire of variant amino acids never found at each position in the OCE clade was built, and mutations were introduced randomly in the ancestral sequence as previously. The structure corresponding to each mutated sequence was predicted using Alpha-Fold2, and structures were compared using Dalilite5 and TMalign. The structure divergence effect $SD_i$ at position i was defined as the average $Z_{\text{(ancestor against mutant i)}} - Z_{\text{(ancestor against itself)}}$ for alanine and deletion mutants targeting position i. The SD compensatory effect at position i was defined as $SD_i - [1/\text{number of mutations} * SD_{\text{(shuffle or new mutant including mutation at position i)}}]$ mean-normalized with SD distributions. The net stabilization factor is the difference between SD and SD compensatory effect at position i.

### Reconstruction of structure and frustration evolutionary paths

Clusters of related OCEs among homologs from NCBI were identified using MMseqs2 (as mentioned previously) and redundant sequences (>97% identity) were filtered out with Cdhit[94]. Signal peptides were removed based on predictions from signalP version 6.0. For these analyses, a selection of the largest alignable groups were retained for six families. Alignments of these clusters were generated with Decipher using default parameters[91]. Alignments were manually curated using JalView version 2.11.2.5 to remove sequences that had poorly supported InDels and the N terminus was trimmed to the same location for all cluster members following the signal peptide. Phylogenetic trees were built from these alignments using phyML 3.1 using the approximate likelihood ratio test for branch support, 4 categories of substitution rates and an estimated gamma distribution parameter[95]. Trees were time-calibrated using mid-point rooting and root maximum age calibration with the chronos function of the R package ape v. 5.6-2[96], with maximum ages inferred using TimeTree 5[97]. Ancestral OCE structures were determined using GRASP and AlphaFold2 as described above. Residue frustration was calculated on Alphafold rank 1 models using frustratometeR version 0.1.0[58] with long-range electrostatics disabled, mutational calculation mode and sequence separation of 3. Residues were considered highly frustrated for a frustration index (FI) lower than −1. FI variance in OCE clades was calculated as the variance of FIs at a given aligned position for all aligned OCEs. To determine frustration variation along tree branches, the difference between FI in the ancestral protein and FI in the descendant protein was calculated at each aligned position (ΔFI), and ΔFIs summed up along the alignment (ΣΔFI). Structure comparisons were performed using DALILite 5.0 as described above. Structural divergence along branches corresponded to the difference between Dali Z-score for comparison of ancestral structure with itself and ancestral structure with descendent structure. These values were divided by the difference between estimated divergence time of the ancestral and descendent nodes. Trees were rendered using R functions from the package phytools[98].

### Statistics and reproducibility

Statistical analyses were performed with RStudio v1.4.1106 and R x64 4.0.4. Data distributions were modeled using LOESS local regression with confidence or prediction intervals shown. Differences of quantitative data between groups were calculated using 2-sided unpaired t-test. Null probability of co-mutation occurrence was determined by shuffling extant variable residues 10,000 times. P-values were calculated using Bonferroni correction for multiple testing. For mutational scans, all positions were mutated to alanine or deleted, sets of 100 random sequences with multiple mutations were analyzed to reach a sample size comparable to modern, alanine scan and deletion scan sets. Random multiple mutants were designed by assigning randomly amino-acids independently to each position in proteins. For family-wise analyses, the results from at least two unrelated families spanning >15 species and >65 proteins are reported. Results reported in the manuscript were successfully replicated on all protein families analyzed, with statistics reported in the Source data File.

### Reporting summary

Further information on research design is available in the Nature Portfolio Reporting Summary linked to this article.

### Data availability

Source data on zenodo.org under https://doi.org/10.5281/zenodo.7506581[99]. Raw data with accession codes and protein identifiers are listed in Source data file 1. Source data are provided with this paper.

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

## Acknowledgements

SR was supported by the French Laboratory of Excellence project 'TULIP' (ANR-10-LABX-41; ANR-11-IDEX-0002-02), l'Agence Nationale pour la Recherche (ANR-19-CE20-15, ANR-21-CE20-10, ANR-21-CE20-30), and the INRAE. S.R. is grateful the LIPME bioinformatics team and especially Ludovic Legrand for providing help and computing and storage resources. MCD was supported by a co-investment between the Grains Research and Development Corporation of Australia and Curtin University on grant CUR00023. MCD would like to acknowledge that this work was supported by resourced provided by the Pawsey Supercomputing Research Centre with funding from the Australian Government and the Government of Western Australia.

## Author contributions

S.R. conceptualized the study. M.C.D. and S.R. designed, performed and interpreted analyses, wrote and edited the manuscript.

## Competing interests

The authors declare no competing interests.
