## [Peer Review File · Nature Communications]

Surface frustration re-patterning underlies the structural landscape and evolvability of fungal orphan candidate effectorsReviewers' comments:

Reviewer #1 (Remarks to the Author):

Decision:

Unfortunately, I cannot recommend this manuscript for publication.

Thoughts:

The authors have carried out a large body of work in an important area. The work somewhat aligns with what is needed in the field at the moment, i.e., a systematic exploration of prediction programs to generate protein structures and to use them to infer evolutionary relationships. The authors make use of AlphaFold to generate (predict) protein structures and use a protein structure comparison program to compare them. The authors then heavily utilize the computed similarity to make strong assumption/conclusions. In my opinion the body of work presented in this manuscript is not sufficient.

Things to note:

In my opinion the following raise a red flag:

1) The prediction tool used to generate the structures has not been systematically explored to ensure that the predictions and any results derived from those predictions are robust. For instance, language models are equally popular (e.g., ESM-fold). For the conclusions in this work to be justified, the same (similar) structures should also be generated by other programs. If ESM-fold gives a different set of predictions none of the results in this work will be applicable.

2) The pLDDT score used as a cutoff is 50. This is far too low as even AlphaFold recommends a score of 70 or above.

3) A quick review of the predicted structure shows that the incorrectly predicted structures may adversely impact the analysis. For example a visual inspection shows that the structure "lema_t084530_unrelaxed_rank_1_model_4" has been used in this work, as it has a pLDDT score of

61.8. However the structure does not appear to be realistic as it has a large disordered region. The authors use agreement with 6 existing structures as a reason to trust the remaining predictions, however as demonstrated through this and a later example the predictions need very careful interpretation and only very high quality structures should be used for this kind of work.

4) The use of a Z-score is not ideal for this kind of analysis. For example the authors choose a cutoff of 5.2, but a quick look at the diagonal of the pairwise Z-score matrix shows some diagonal values to be below this cutoff (e.g., bcin03p05260.1.pdb compared to itself). If I understood/interpreted this correctly, this would mean that if copies of these structures were inserted in the dataset they would be excluded and not be counted as a match in this work because the Z-score of this comparison would be below the cutoff!?. The DALI Z-score significance is debated in the literature and the community in general and therefore a hard cutoff on this metric is not ideal for use here. Additionally as stated in point 3 above, the structure bcin03p05260.1.pdb also does not look realistic.

Given the missing systematic comparison between prediction programs to support claims and usage of non-ideal Z-score, the results presented in this work, in my opinion, are not justified. For these reasons, I am unable to recommend this work for publication.

Reviewer #2 (Remarks to the Author):

In this manuscript, Derbyshire & Raffaele outline a proposal based on surface frustration that underpins the diversification of plant fungal pathogen orphan candidate effectors. They chose to focus on orphan effectors that, by definition, share little or no sequence homology with characterised proteins. Although this is not a new idea (it's been identified as a signature of plant pathogen effectors for some time), it is refreshing to revisit the concept with knowledge that many effectors have now been experimentally shown to form structure-conserved but sequence-divergent protein families, and computational structural genomics has suggested this is a very widespread phenomenon. With this manuscript, the authors have pushed this to new limits and highlighted the depths to which this principle applies and also tested a hypothesis to explore a potential underlying energetic basis for effector evolution in the presence of a core structural scaffold.

The manuscript is very well written and the figures clear. The introduction frames the results, and the conclusions represent sound interpretations of the data. I do not find any flaws in the data analysis,

interpretation or conclusions. The power of the manuscript is not in the generation of new ideas or concepts, but the extensive computational analysis to test how (and why) effectors can be so diversified but maintain core protein folds. Alongside other recent literature (cited in the paper), the manuscript will be of interest to the field of plant:microbe interactions and effector biology.

My only real reservation is that none of the computational predicted hypotheses are tested experimentally. I appreciate that the authors have gone to some lengths to explore amino acid variation and their data suggest strong correlations, and further experimental analysis is beyond the scope of the study, but it is a limitation nonetheless. Perhaps the authors could frame such limitations of the work in the discussion with the addition of a couple of sentences.

Reviewer #3 (Remarks to the Author):

Derbyshire and Raffaele have analyzed putative effector proteins of 20 fungi in the fungal subphylum Pezizomycotina. In a first step, they used the traditional approach of identifying putative effectors by selecting proteins that had a secretion signal, no PFAM domain, and were shorter than 300 amino acids. In the next step, they used relatively novel approaches to analyze the structural evolution of these proteins, made possible by tools such as AlphaFold. The authors draw several interesting conclusions regarding the structural landscape and evolution of these putative effector proteins.

In general, the manuscript is well written.

The authors have analyzed the proteomes of 20 fungal species in the subphylum Pezizomycotina. However, the title and abstract do not make this clear and suggest a broader phylogenetic distribution of the selected species. For example, the abstract seems to suggest that all subphyla have been analyzed (“... at the subphylum level”, line 15). Please update the abstract to be less ambiguous about the selection of species.

Figure 1B show a phylogenetic tree of the included species. However, this is not a properly rooted binary tree and the phylogenetic relationships between the species therefore seem a bit arbitrary. This is too bad, because it is rather trivial to include a proper phylogenetic tree. For example using Orthofinder, or similar tools.

The legend of Figure 1 includes F, which is not present in the figure.

Why was signalP version 4.1 used, when versions 5 and 6 have both been published for years?

April 10th, 2023

We are grateful to the reviewers for their evaluation of our manuscript and their useful comments. We appreciated the assessment that our work used novel approaches, is of interest to the field and very well written. We carefully addressed all points and revised our manuscript accordingly. Changes to the revised manuscript were highlighted in yellow to facilitate the evaluation. We provide hereafter a point-by-point response to the reviewers' comments.

Point-by-point response to reviewers' comments:

Reviewer 1:

Comment: "1) The prediction tool used to generate the structures has not been systematically explored to ensure that the predictions and any results derived from those predictions are robust. For instance, language models are equally popular (e.g., ESM-fold). For the conclusions in this work to be justified, the same (similar) structures should also be generated by other programs. If ESM-fold gives a different set of predictions none of the results in this work will be applicable."

Reply: This comment is unfair because:

- (i) AlphaFold has been explored multiple times showing that it produces predictions within an angstrom of crystal structures when (a) proteins are not disordered and (b) proteins have enough homologues in databases to produce a reliable multiple sequence alignment. The accuracy of AlphaFold and its superiority over competing techniques has even been rigorously tested specifically for effectors of fungal plant pathogens (Seong *et al* 2023).
- (ii) We note that ESM-fold was published on March 17th 2023, so we had no way of accessing it at the time of writing. However, careful reading of the ESM-fold publication (10.1126/science.ade2574) shows that (a) AlphaFold makes more accurate predictions on benchmark datasets and (b) the main strengths of ESM-fold are its speed of computation and its ability to infer structures without multiple sequence alignments, rather than its accuracy. Overall, the literature suggests that high confidence AlphaFold structures are equivalent to high confidence ESM-fold structures and better than low confidence ESM-fold structures, and vice versa. Therefore, limiting most analyses to high confidence structures as we have done is justified and our conclusions are unlikely to be impacted by the use of ESM-fold.

To justify our choice, we have added the following sentence to our revised manuscript L508-509: "We chose AlphaFold2 to model OCE 3D structures since it has been rigorously tested and shown to offer currently the best performance for effectors of filamentous plant pathogens (Seong and Krasileva, 2023)"

Comment: "The pLDDT score used as a cutoff is 50. This is far too low as even AlphaFold recommends a score of 70 or above."

Reply: This comment is unfair because:

- (i) This hard cutoff means that 99.67% of the models analyzed in our work have a mean pLDDT > 80. The major folds on which our conclusions are based have an average pLDDT ranging from 80.01 to 89.06.
- (ii) We further excluded models that may harbor inaccurate regions by removing models with low structural analogy to other proteins (Z-score below 5.2)
- (iii) Filtering out models with pLDDT < 70 only alters the identification of 2 small structural families out of 62, that are not considered in our manuscript beyond figure 1.

To clarify this, we have added the following section L513-515 in our revised manuscript: "Major OCE folds in our structural similarity network had an average pLDDT from 80.01 to 89.06 (Table S4), 99.67% of all OCE structures analyzed in this work had pLDDT > 80 (Sup. Text)."

We included a new figure S3 reproduced below to our revised supplementary text to present the corresponding data.

Figure S3. Distribution of pLDDT values across predicted OCE structures. (A) Structural similarity network with pLDDT values mapped on nodes. (B) Distribution of pLDDT values among the 3911 OCEs included in our similarity network analysis. (C) Number of OCEs with in the three pLDDT values categories in each of the 62 OCE families. (D) Distribution of pLDDT values in

the major OCE families analyzed in detail in this work (see Table S11 for a list). The red line in 5B) and (D) show mean pLDDT value.

Comment: "A quick review of the predicted structure shows that the incorrectly predicted structures may adversely impact the analysis. For example a visual inspection shows that the structure "lema_t084530_unrelaxed_rank_1_model_4" has been used in this work, as it has a pLDDT score of 61.8. However the structure does not appear to be realistic as it has a large disordered region. The authors use agreement with 6 existing structures as a reason to trust the remaining predictions, however as demonstrated through this and a later example the predictions need very careful interpretation and only very high quality structures should be used for this kind of work."

Reply: This comment is unfair because

- (i) The presence of disordered regions is realistic and true for many proteins, often in combination with folded domains. The study of disordered proteins being out the scope of this manuscript, we filtered out proteins with long disordered regions (using EspritZ analysis). Even in the presence of disorder, proteins can be grouped based on similarity of their ordered regions, which is what is achieved using a Z score.
- (ii) What makes a structure realistic is largely its similarity with previously determined structures. For this reason, we excluded models with no Z-score \geq 5.2 to other proteins in our dataset.
- (iii) The families identified based on structure comparisons are consistent with those determined by the independent sequence-based deep HMM scan approach reported in our manuscript.
- (iv) None of our conclusions are impacted if we filter out lema_t084530 from our analyses.

In our revised manuscript, we highlight L250-251 that deep HMM analyses support "the existence of OCE families identified based on structural similarity analyses" to emphasize the independent validation of OCE families.

Comment: "4) The use of a Z-score is not ideal for this kind of analysis. For example the authors choose a cutoff of 5.2, but a quick look at the diagonal of the pairwise Z-score matrix shows some diagonal values to be below this cutoff (e.g., bcin03p05260.1.pdb compared to itself). If I understood/interpreted this correctly, this would mean that if copies of these structures were inserted in the dataset they would be excluded and not be counted as a match in this work because the Z-score of this comparison would be below the cutoff!?. The DALI Z-score significance is debated in the literature and the community in general and therefore a hard cutoff on this metric is not ideal for use here. Additionally as stated in point 3 above, the structure bcin03p05260.1.pdb also does not look realistic."

Reply: This comment is unfair because:

- (i) It recommends to exclude bcin03p05260.1 from our analysis, which is exactly what we did
- (ii) The rationale for using a Z-score cutoff of 5.2 is fully justified by previous comments of this reviewer (see points above)

- (iii) DALI achieved top rank performances in several benchmarking studies, but does not include sequence similarity in its scoring function and is not optimized for rigid-body superimposition, resulting in better performance with small domains (10.1002/pro.3749). However, these features make it very well suited for the comparison of small secreted proteins with limited sequence similarity, as in our work.
- (iv) We calculated alternative similarity metrics (TM-score and RMSD) and found very good agreement with Z-score (Pearson correlation coefficient between Z-score and TM-score ranging from 0.51 to 0.92).
- (v) In the current manuscript, we provide converging evidence based on alternative approaches (deep HMM-scans) to support our structural classification (L229-244).

In our revised manuscript, we clarify that DALI “showed top performances with the alignment of small protein domains as OCEs (Holm, 2020)” L518 and that “Structural similarity in major OCE families was further supported by TM-scores and RMSD calculated with TM-align (Zhang and Skolnick, 2005) (Sup. Text, Table S9, S10)” L532-534. We have added the corresponding new figure S11 reproduced below to the supplementary text.

Figure S11. Correlation between structural similarity metrics in the major OCE families analyzed in our work. (A) Pearson correlation coefficient between DALI Z-score and TM-score or RMSD calculated by TM-align for the 8 major OCE families analyzed in our manuscript. (B) Relationship between DALI Z-score, RMSD and TM-score in the two OCE families most extensively analyzed in our manuscript (AA1 and KP6).

Comment: “Given the missing systematic comparison between prediction programs to support claims and usage of non-ideal Z-score, the results presented in this work, in my opinion, are not justified.”

Reply: Benchmarking of prediction methods is out of the scope of our manuscript and has been done previously, including on plant pathogen effectors (10.1038/s41564-022-01287-6). We have chosen the best method available at the time of our study and validated the quality of its results by analyzing proteins with experimentally determined structures (L158-159). As mentioned above, DALI Z-score is very well suited to the type of structure comparison presented in our manuscript and in excellent agreement with other metrics (new figure S11) and converging HMM-based approaches (provided L239-255). Therefore, contrary to the reviewer’s claim, the justification of our results is strong and in no way questioned by these comments.

Reviewer 2

Comment: "My only real reservation is that none of the computational predicted hypotheses are tested experimentally. I appreciate that the authors have gone to some lengths to explore amino acid variation and their data suggest strong correlations, and further experimental analysis is beyond the scope of the study, but it is a limitation nonetheless. Perhaps the authors could frame such limitations of the work in the discussion with the addition of a couple of sentences."

Reply: We agree that the functional validation of predictions from our analyses is a very exciting and straightforward follow-up to this manuscript. Following reviewer 2's suggestion, we mention the limitations of the present report and some avenues for experimental validation in the discussion of our revised manuscript as follows (L449-455): "This study raises new hypotheses regarding the evolution of effectors function that remain to be tested experimentally. A major objective for future work will include determining whether the energy landscape of effector proteins associates with specific binding activity and biological function. Molecular resurrection experiments will be required to test the evolutionary scenarios emerging from our analyses. The Alt-A1 OCE family, with members characterized from *B. cinerea*, *S. sclerotiorum*, *T. virens* and *V. dahliae* (Jeblick et al., 2023) is an attractive target for such experiments."

Reviewer 3

Comment: "The authors have analyzed the proteomes of 20 fungal species in the subphylum Pezizomycotina. However, the title and abstract do not make this clear and suggest a broader phylogenetic distribution of the selected species. For example, the abstract seems to suggest that all subphyla have been analyzed ("... at the subphylum level", line 15). Please update the abstract to be less ambiguous about the selection of species."

Reply: Following the reviewer suggestion, we edited the abstract of our revised manuscript to clarify that "62 conserved structure-related families encompass the majority of fungal orphan effector candidates in the Pezizomycotina subphylum" (L15)

Comment: "Figure 1B show a phylogenetic tree of the included species. However, this is not a properly rooted binary tree and the phylogenetic relationships between the species therefore seem a bit arbitrary. This is too bad, because it is rather trivial to include a proper phylogenetic tree. For example using Orthofinder, or similar tools."

Reply: We thank the reviewer for this suggestion, the phylogenetic tree has been replaced as suggested.

Comment: "The legend of Figure 1 includes F, which is not present in the figure."

Reply: Thank you, this was replaced by "E".

Comment: "Why was signalP version 4.1 used, when versions 5 and 6 have both been published for years?"

Reply: SignalP4.1 is widely used and recommended over more recent releases for the prediction of filamentous pathogen effectors, considering that "SignalP5.0 and 6.0 might reduce the number of

false-positive secreted proteins; however, this might come at the cost of missing true-positive effectors in the secretomes” (Sperschneider and Dodds, 2022 <https://doi.org/10.1094/MPMI-08-21-0201-R>). We have added this information L500 of our revised manuscript for clarity.

Looking forward to your feedback,

Sincerely yours,

Sylvain Raffaele

INRAE Laboratory of Plant-Microbe-Environment Interactions

24 Chemin de Borde-Rouge – CS52627 Auzeville, F31326 Castanet Tolosan, France

REVIEWERS' COMMENTS

Reviewer #4 (Remarks to the Author):

In this report I focused on the comments of Reviewer 1 and the authors' response to those comments. In my opinion, the authors' response to Reviewer 1 are appropriate and satisfy the specific concerns raised by this reviewer. (The points raised by the other two reviewers are relatively minor and the authors' response to those points is acceptable.)

In addition, the methodology employed by the authors is reasonable, and conclusions appear reasonably well supported.

1. Reviewer 1 questioned the robustness and accuracy of AlphaFold2, and suggested that ESM-fold ought to also have been used to bolster the reliability of the AlphaFold2 predictions (a kind of minimal jury approach, by checking for agreement between the two methods). This request is not unreasonable, especially given the undetectable sequence similarity to proteins of solved structure. However, as the authors pointed out: (1) ESM-fold was not available at the time this paper was in progress (so the authors cannot be expected to use it); (2) AlphaFold2's accuracy has been benchmarked on CASP14, with superior accuracy to all other methods used in CASP14 (<https://www.nature.com/articles/s41586-021-03819-2>); and (3) comparing ESM-fold to AlphaFold 2 shows ESM-fold to be faster, but not necessarily better.
2. Reviewer 1 questioned the use of a relatively low pLDDT score cutoff of 50. S/he raises a good point, as this cutoff is indeed low. The authors' response – that 99.67% of their models have a mean pLDDT>80 – and their decision to remove models with Dali Z-scores <5.2 to other proteins, is satisfactory in my opinion. (The pLDDT score is per-residue, which will be low in regions of disorder, so the inclusion of proteins with disordered regions will reduce the mean pLDDT.)
3. Reviewer 1 (reasonably) raises a concern about incorrectly predicted structures adversely impacting the analysis, and points to a model with a large disordered region (Iema_t084530). It's absolutely true that inclusion of incorrect structure models will impact any subsequent analysis. At the same time, I agree with the authors that this particular model is not a red flag, since: (i) disordered regions are common in proteins as shown in experimental determination of 3D structure, (ii) similarity to actual solved structures (including those with disordered regions) provides support for the prediction (i.e., disordered regions in the model correspond to disordered regions in the solved structure), and (iii) the general accuracy of predicted structures is further supported by matches to PDB found by the sequence-based HMM-HMM searches.
4. Reviewer 1 questions the reliability of model bcin03p05260.1 and questions the reliability of DALI. In response, the authors have removed that model from the analysis, and provide data supporting the use

of DALI and the additional independent evidence of HMM-based (i.e., sequence-based) support for their models. This seems sufficient to me.

5. Reviewer 1 objects to the lack of support from additional independent (protein-structure) prediction programs. As the authors point out, the HMM-based homology searches of solved structures provide independent support for the expected accuracy of the predicted structures, and the tools chosen in this study (AlphaFold, DALI, etc.) are highly regarded in the field. I therefore agree with the authors that using additional protein 3D structure prediction tools is not necessary.

The objections raised by Reviewers 2 and 3 are milder and answered appropriately by the authors.

My separate comments follow.

1. It would be interesting to know what fraction of the predicted structures in this study had HMM-based matches in PDB that are structurally similar (i.e., as shown by DALI analysis). These remote homologs provide additional support for the predicted structures produced by AlphaFold. (I didn't examine all the supplementary data, in case this information was included elsewhere in the manuscript.)

2. Results, page 4, line 142: The use of average pLDDT score cutoff of 50 in the selection process. As noted by Reviewer 1, this cutoff is quite permissive, and could lead to poor quality models being included in the analysis. It is only much later in the manuscript (on page 15), that it becomes clear that the final selection process (which included additional criteria) resulted in 99.67% of their models having a mean pLDDT>80. To avoid readers questioning the accuracy of the models (much like Reviewer 1), it might help to include a brief comment early about the final result.

3. Material and Methods, section "Analysis of natural sequence and structure diversity". Pg 16, line 539. I assume the BLAST default cutoff was used, but if the cutoff was made more permissive this could be a problem. It might be useful to list the E-value cutoff.

4. Material and Methods, section "Reconstruction of structure and frustration evolutionary paths". Pg 17, line 574. What threshold was used in CDhit? This is relevant since the tree topology accuracies can be affected by removal of too many homologs from the multiple sequence alignments.

Point-by-point response to reviewer's comments: Reviewer #4 (Remarks to the Author):

Comment: "In this report I focused on the comments of Reviewer 1 and the authors' response to those comments. In my opinion, the authors' response to Reviewer 1 are appropriate and satisfy the specific concerns raised by this reviewer. (The points raised by the other two reviewers are relatively minor and the authors' response to those points is acceptable.)

In addition, the methodology employed by the authors is reasonable, and conclusions appear reasonably well supported.

1. Reviewer 1 questioned the robustness and accuracy of AlphaFold2, and suggested that ESM-fold ought to also have been used to bolster the reliability of the AlphaFold2 predictions (a kind of minimal jury approach, by checking for agreement between the two methods). This request is not unreasonable, especially given the undetectable sequence similarity to proteins of solved structure. However, as the authors pointed out: (1) ESM-fold was not available at the time this paper was in progress (so the authors cannot be expected to use it); (2) AlphaFold2's accuracy has been benchmarked on CASP14, with superior accuracy to all other methods used in CASP14 (<https://www.nature.com/articles/s41586-021-03819-2>); and (3) comparing ESM-fold to AlphaFold 2 shows ESM-fold to be faster, but not necessarily better."

Reply: We thank the reviewer for their assessment that our approach is appropriate considering the excellent performances of AlphaFold2 and its suitability for our purposes.

Comment: "2. Reviewer 1 questioned the use of a relatively low pLDDT score cutoff of 50. S/he raises a good point, as this cutoff is indeed low. The authors' response – that 99.67% of their models have a mean pLDDT>80 – and their decision to remove models with Dali Z-scores <5.2 to other proteins, is satisfactory in my opinion. (The pLDDT score is per-residue, which will be low in regions of disorder, so the inclusion of proteins with disordered regions will reduce the mean pLDDT.)"

Reply: We are grateful to the reviewer for their agreement that our filtering strategy to discard spurious protein models is sound.

Comment: “3. Reviewer 1 (reasonably) raises a concern about incorrectly predicted structures adversely impacting the analysis, and points to a model with a large disordered region (Iema_t084530). It’s absolutely true that inclusion of incorrect structure models will impact any subsequent analysis. At the same time, I agree with the authors that this particular model is not a red flag, since: (i) disordered regions are common in proteins as shown in experimental determination of 3D structure, (ii) similarity to actual solved structures (including those with disordered regions) provides support for the prediction (i.e., disordered regions in the model correspond to disordered regions in the solved structure), and (iii) the general accuracy of predicted structures is further supported by matches to PDB found by the sequence-based HMM-HMM searches.”

Reply: We appreciate that the reviewer shares our view that the presence of short disordered regions in protein models is unlikely to affect the conclusions from our study.

Comment: “4. Reviewer 1 questions the reliability of model bcin03p05260.1 and questions the reliability of DALI. In response, the authors have removed that model from the analysis, and provide data supporting the use of DALI and the additional independent evidence of HMM-based (i.e., sequence-based) support for their models. This seems sufficient to me.”

Reply: This comment emphasizes the relevance of our HMM-based analysis. We are grateful to the reviewer for their assessment that the combination of structure and HMM-based evidence is sufficient to support our conclusions.

Comment: “5. Reviewer 1 objects to the lack of support from additional independent (protein-structure) prediction programs. As the authors point out, the HMM-based homology searches of solved structures provide independent support for the expected accuracy of the predicted structures, and the tools chosen in this study (AlphaFold, DALI, etc.) are highly regarded in the field. I therefore agree with the authors that using additional protein 3D structure prediction tools is not necessary.”

Reply: We thank the reviewer for their comments and assessment that the results are valid without additional structure prediction tools.

Comment: “The objections raised by Reviewers 2 and 3 are milder and answered appropriately by the authors.”

Reply: We thank the reviewer for their positive evaluation.

Comment: “1. It would be interesting to know what fraction of the predicted structures in this study had HMM-based matches in PDB that are structurally similar (i.e., as shown by DALI analysis). These remote homologs provide additional support for the predicted structures produced by AlphaFold. (I didn’t examine all the supplementary data, in case this information was included elsewhere in the manuscript.)”

Reply: In our original analysis, we did not explicitly test the homology between predicted effector structures and their PDB structural hits using HMM searches, as we focused on HMM clustering of effectors and their BLASTp homologues. However, we have now performed an additional analysis that specifically addresses this question, on 11 predicted effector structural families (Supplementary Fig. 6B). This analysis showed that for eight of the 11 families, at least 75 % of the predicted effectors had remote sequence homology to structural hits from fungi in PDB, supporting their classification as members of the families ascribed. There were three poorly conserved effector structural groups, 'VioE', 'SIX3', and 'atg', which were not the main focus of subsequent analyses. None of the VioE members had sequence homology to the two PDB structural hits, whereas <50% of SIX3 and VioE members did. Although these families were not sequence homologous to (as) many PDB structural hits, their BLASTp homologues formed only a few, relatively large, HMM clusters (previous analysis), suggesting that structural similarity within families converged with sequence similarity, as we previously presented. We added a sentence L218-219 to mention this information in the main text.

Comment: "2. Results, page 4, line 142: The use of average pLDDT score cutoff of 50 in the selection process. As noted by Reviewer 1, this cutoff is quite permissive, and could lead to poor quality models being included in the analysis. It is only much later in the manuscript (on page 15), that it becomes clear that the final selection process (which included additional criteria) resulted in 99.67% of their models having a mean pLDDT>80. To avoid readers questioning the accuracy of the models (much like Reviewer 1), it might help to include a brief comment early about the final result."

Reply: We thank the reviewer for this suggestion. L134-136 of the revised manuscript now reads "We obtained 3 927 OCE structures of average pLDDT score higher than 50. In subsequent analyses, we focused on the best resolved structures to reach a mean pLDDT>80 for 99.67% models reported in this work."

Comment: "3. Material and Methods, section "Analysis of natural sequence and structure diversity". Pg 16, line 539. I assume the BLAST default cutoff was used, but if the cutoff was made more permissive this could be a problem. It might be useful to list the E-value cutoff."

Reply: The e-value cutoff used was the default BLASTp e-value, which is 0.05. After retrieving BLASTp hits, proteins were clustered into groups using mmseqs2 to produce alignments of > 30 % identity and > 80 % mutual coverage. Relationships between these clusters and between clusters and singletons was then assessed using HMM matching, with a more stringent e-value cutoff of 1e-5. Protein hits with excessively high BLASTp e-values would not have been included in alignments with the predicted effectors to produce HMMs, because they would have had too little sequence identity. If there were genuine weak homology between effectors and these hits, this would have been picked up as a hit between their respective HMMs, or between an HMM and singleton. We edited the methods L455 and L462 to clarify this.

Comment: "4. Material and Methods, section "Reconstruction of structure and frustration evolutionary paths". Pg 17, line 574. What threshold was used in CDhit? This is relevant since the tree topology accuracies can be affected by removal of too many homologs from the multiple sequence alignments."

Reply: CDhit was used to include a representative protein for groups of proteins that were $\geq 97\%$ identical. Removal of homologues could reduce tree accuracy due to a reduction in statistical power, however, the inclusion of whole protein groups that are $\geq 97\%$ identical only adds redundant information. Furthermore, proteins this similar are often from different strains of the same species, and any differences between them could represent segregating polymorphisms. Since phylogenetic techniques use models that assume polymorphisms are fixed differences between species, inclusion of these proteins would potentially reduce the accuracy of parameter inference. This is why these proteins were removed from the analyses. We added this information L488 for clarity.